# Oral Supplementation of the Vitamin D Metabolite 25(OH)D_3_ Against Influenza Virus Infection in Mice

**DOI:** 10.3390/nu12072000

**Published:** 2020-07-05

**Authors:** Hirotaka Hayashi, Masatoshi Okamatsu, Honami Ogasawara, Naoko Tsugawa, Norikazu Isoda, Keita Matsuno, Yoshihiro Sakoda

**Affiliations:** 1Laboratory of Microbiology, Faculty of Veterinary Medicine, Hokkaido University, Kita 18 Nishi 9, Kita-ku, Sapporo 060-0818, Japan; hayashihirotaka@vetmed.hokudai.ac.jp (H.H.); okamatsu@vetmed.hokudai.ac.jp (M.O.); matsuno@vetmed.hokudai.ac.jp (K.M.); 2Department of Health and Nutrition, Faculty of Health and Nutrition, Osaka Shoin Woman’s University, Hishiyanishi 4-2-26, Higashiosaka 577-8550, Japan; ogasawara.honami@osaka-shoin.ac.jp (H.O.); tsugawa.naoko@osaka-shoin.ac.jp (N.T.); 3Unit of Risk Analysis and Management, Research Center for Zoonosis Control, Hokkaido University, Kita 20 Nishi 10, Kita-ku, Sapporo 001-0020, Japan; isoda@czc.hokudai.ac.jp; 4Global Station of Zoonosis Control, Global Institution for Collaborative Research and Education (GI-CoRE), Hokkaido University, Sapporo 001-0020, Japan

**Keywords:** vitamin D, 25-hydroxyvitamin D_3_, cytokines, influenza

## Abstract

Vitamin D is a fat-soluble vitamin that is metabolized by the liver into 25-hydroxyvitamin D [25(OH)D] and then by the kidney into 1,25-dihydroxyvitamin D [1,25(OH)_2_D], which activates the vitamin D receptor expressed in various cells, including immune cells, for an overall immunostimulatory effect. Here, to investigate whether oral supplementation of 25-hydroxyvitamin D_3_ [25(OH)D_3_], a major form of vitamin D metabolite 25(OH)D, has a prophylactic effect on influenza A virus infection, mice were fed a diet containing a high dose of 25(OH)D_3_ and were challenged with the influenza virus. In the lungs of 25(OH)D_3_-fed mice, the viral titers were significantly lower than in the lungs of standardly fed mice. Additionally, the proinflammatory cytokines IL-5 and IFN-γ were significantly downregulated after viral infection in 25(OH)D_3_-fed mice, while anti-inflammatory cytokines were not significantly upregulated. These results indicate that 25(OH)D_3_ suppresses the production of inflammatory cytokines and reduces virus replication and clinical manifestations of influenza virus infection in a mouse model.

## 1. Introduction

Vitamin D, primally contained in oily fishes, is a fat-soluble vitamin that promotes bone formation and calcium absorption [1]. In addition to being synthesized from cholesterol by reacting with ultraviolet rays in the epidermis, vitamin D can be ingested and is absorbed in the small intestine. It is metabolized in the liver to become highly stable 25-hydroxyvitamin D [25(OH)D], which circulates in the blood. The kidney metabolizes 25(OH)D into 1,25-dihydroxyvitamin D [1,25(OH)_2_D], which binds to the vitamin D receptor (VDR) expressed in various cells [2]. The metabolite 1,25(OH)_2_D promotes the regeneration of respiratory epithelial cells [3] and has also been implicated in the regulation of inflammation by binding to the VDR on immune cells such as dendritic cells, macrophages, and T cells [4]. Activation of the VDR by 1,25(OH)_2_D suppresses the production of inflammatory cytokines such as tumor necrosis factor (TNF)-α and interleukin (IL)-5 by regulating target genes of T cells and macrophages and reducing inflammation [5]. It also directly interacts with B cells and T cells [6]. Furthermore, 1,25(OH)_2_D reduces inflammatory cytokines in challenge studies with the pertussis toxin [7], *Mycobacterium tuberculosis* [8], and poliovirus infection [9].

The influenza A virus is an enveloped, single-stranded, segmented, negative-sense RNA virus that infects the upper respiratory tract in humans, causing influenza [10]. Influenza A virus infection causes severe seasonal epidemics especially in the elderly and in young children, partly because of poor immune function. Reduction of the serum vitamin D metabolite 25(OH)D, decreases in endogenous vitamin D synthesis, and poor dietary vitamin D intake can also weaken the immune system [11]. Furthermore, vitamin D has been suggested as having immunostimulatory effects against influenza [12]. A major form of 25(OH)D, 25-hydroxyvitamin D_3_ [25(OH)D_3_], is absorbed efficiently in the small intestine [13] and has a relatively long half-life of 15 days [14]. In the present study, we used a mouse model of influenza virus infection and direct oral ingestion of 25(OH)D_3_ to evaluate the effect of this metabolite on the clinical manifestations of influenza virus infection, virus replication, and cytokine levels.

## 2. Materials and Methods

### 2.1. Virus and Cells

Influenza virus A/Puerto Rico/8/1934 (PR8) (H1N1) was propagated in 10-day-old embryonated chicken eggs at 35 °C for 48 h, and the collected allantoic fluid was stored at −80 °C until use. Madin–Darby canine kidney (MDCK) cells were maintained in minimum essential medium (MEM) (Nissui Pharmaceutical, Tokyo, Japan) supplemented with 10% nonimmobilized fetal calf serum (FCS; SAFC Biosciences, Lenexa, KS, USA), 0.3 mg/mL L-glutamine (Wako Chemicals, Tokyo, Japan), 100 U/mL penicillin G (Meiji Seika Pharma, Tokyo, Japan), 0.1 mg/mL streptomycin (Meiji Seika Pharma, Tokyo, Japan), and 8 μg/mL gentamicin (Takata Pharmaceutical, Saitama, Japan) in an incubator at 37 °C with 5% CO_2_.

### 2.2. Animals and Diet

Seven-week-old female BALB/c mice (Japan SLC, Shizuoka, Japan) used in the present study were acclimatized for one week, divided into two groups (*n* = 18 mice/group), and fed different diets, i.e., AIN-93G standard diet (vitamin D_3_ 0.25 mg/100 g) or AIN-93G with 25(OH)D_3_ [125 mg/kg (125 ppm)] (Oriental Yeast, Tokyo, Japan). Body weights and clinical signs were observed for 7 weeks after beginning the different diets.

### 2.3. Measurement of 25(OH)D_3_ and 24,25(OH)_2_D_3_

Plasma concentrations of 25(OH)D_3_ and 24-hydroxyvitamin D_3_ [24,25(OH)_2_D_3_] were measured using a modified method of liquid chromatography–atmospheric pressure chemical ionization–mass spectrometry (LC–APCI–MS)/MS [15]. This method involved the use of deuterated 25(OH)D_3_ (*d*_6_-25[OH]D_3_) as an internal standard compound and the selection of a precursor and a product ion with an MS/MS multiple reaction monitoring (MRM) method. Measuring was conducted as previously described [16]. An API3000 LC/MS/MS System (Applied Biosystems, Foster City, CA, USA) was used. The high-performance liquid chromatography column used was a CAPCELL PAK C18 UG120, 5 μm [4.6 I.D. × 250 mm] (Shiseido, Tokyo, Japan). All MS data were collected in positive-ion mode, and quantitative analysis was carried out using MS/MS–MRM of the precursor/product ion.

### 2.4. Virus Challenge in Mice

Virus challenge in mice was conducted as previously described [17]. After 7 weeks of feeding the mice a standard or 25(OH)D_3_-supplemented diet, PR8 (H1N1) was inoculated intranasally using 10 times the 50% mouse lethal dose (MLD_50_) in 30 µL per mouse under anesthesia. The infectivity titer [10 MLD_50_ = 10^4.6^ 50% egg infective dose (EID_50_)] of the inoculum was adjusted by dilution with phosphate-buffered saline (PBS). For 14 days following inoculation, mice were observed daily for body weight, clinical signs, and survival.

### 2.5. Preparation of Lung Homogenates

At pre-inoculation and 3 or 5 days post-inoculation (dpi), mouse lungs were collected after euthanasia (*n* = 6). Lungs were homogenized with 2 mL of transport medium: MEM containing 10,000 U/mL penicillin G, 10 mg/mL streptomycin, 0.3 mg/mL gentamicin, 250 U/mL Nystatin (Sigma-Aldrich, St. Louis, MO, USA), and 0.5% bovine serum albumin fraction V (Roche, Basel, Switzerland). The homogenized lung tissue was centrifuged for 5 min at 4 °C and 8000 rpm. The supernatant was collected, and all samples were stored at −80 °C until quantification of virus titers and cytokines.

### 2.6. Virus Titration in Mouse Lungs

Plaque assays were performed as previously described [18]. Briefly, tenfold dilutions of virus samples or mouse lung homogenates in MEM without FCS were inoculated onto confluent monolayers of MDCK cells and incubated at 35 °C in a 5% CO_2_ incubator for 1 h. Unbound virus was removed in the supernatant, and the cells were washed with PBS. The cells were then overlaid with MEM containing 5 μg/mL acetylated trypsin (Sigma-Aldrich) and 1% Bacto Agar (Becton, Dickinson and Company, Franklin Lakes, NJ, USA). After incubation for 48 h at 35 °C, the cells were stained with 0.005% neutral red. After incubation for another 24 h at 35 °C, the number of plaques was counted. The number of plaque-forming units (PFU) was calculated as the product of the reciprocal value of the highest virus dilution and the number of plaques in the dilution.

### 2.7. Bio-Plex Assay for the Measurement of Anti-Inflammatory and Proinflammatory Cytokines

The Bio-Plex Pro mouse cytokine Th1/Th2 assay (Bio-Rad, Hercules, CA, USA) was used to quantify cytokines including IL-2, IL-4, IL-5, IL-10, IL-12p70, TNF-α, interferon (IFN)-γ, and granulocyte-macrophage colony-stimulating factor (GM-CSF). Lung homogenate samples were diluted 1:2 with the Bio-Plex sample diluent. Standard dilution was performed using the transport medium. Beads were dispensed into wells of a 96-well plate, and the samples, standards, and blanks were added and vortexed for 30 min. After washing with Wash Buffer, a secondary antibody was added to the wells, and the plate was vortexed for 30 min. The wells were washed again, phycoerythrin-conjugated streptavidin was added, and the plate was vortexed for 10 min. After washing again, Assay Buffer was added, and measurement were carried out by Luminex 200 (Merck, Kenilworth, NJ, USA). The concentrations of cytokines (pg/mL) were determined using the Bio-Plex Manager Software (Bio-Rad).

### 2.8. Statistical Analysis

Student’s *t*-test was used to analyze differences in the concentration of vitamin D metabolites, body weights of mice, virus recovery, and cytokines between the two groups. One-way analysis of variance was used to analyze the difference among multiple groups [19]. Animal survival data were analyzed using a log rank (Mantel–Cox) test. All statistical analyses were performed by R version 3.6.3 (R Core Team, 2020).

### 2.9. Ethics Statement

Animal experiments were approved by the Institutional Animal Care and Use Committee of the Faculty of Veterinary Medicine, Hokkaido University (17-0060), and all experiments were carried out per the guidelines of this committee. All applicable international, national, and/or institutional guidelines for the care and use of animals were followed.

## 3. Results

### 3.1. Monitoring of Adverse Effects in Mice

First, the adverse effects of 25(OH)D**_3_** supplementation were evaluated. The groups of mice were fed standard or 25(OH)D_3_-supplemented diet for 7 weeks, and no significant difference in body weight and clinical signs was observed between the two groups (Figure 1); no significant difference in the intake of diet was found (*p* > 0.05). Additionally, no histopathological difference in the kidneys and hearts were detected after 7 weeks (Appendix A), indicating no adverse effects of 125 ppm 25(OH)D_3_ supplementation in mice.

### 3.2. Blood Concentrations of 25(OH)D_3_ and 24,25(OH)_2_D_3_ in Mice

We confirmed that the titer of 25(OH)D**_3_** was significantly higher in the plasma of 25(OH)D**_3_**-fed mice than in that of standardly fed mice after 4 and 7 weeks (Figure 2). Furthermore, the level of 25(OH)D**_3_** after 7 weeks was 40–50 ng/mL, significantly higher than that after 4 weeks in 25(OH)D**_3_**-fed mice.

The compound 24,25(OH)_2_D_3_ is a metabolite of 25(OH)D**_3_**, and in mice, its concentration increases in proportion to the uptake of 25(OH)D**_3_** into the body [16]. As shown in Figure 2, the titer of 24,25(OH)_2_D_3_ was significantly higher in the plasma of 25(OH)D**_3_**-fed mice than in that of standardly fed mice after 4 and 7 weeks. Additionally, the levels of 24,25(OH)_2_D_3_ in the blood were higher than those of 25(OH)D**_3_** after 4 and 7 weeks of dietary supplementation in the 25(OH)D**_3_**-fed mice.

### 3.3. Immunostimulant Test for 25(OH)D_3_ Against Influenza Virus Infection

We next assessed the antiviral effect of the 25(OH)D**_3_**-supplemented diet (125 ppm) against influenza virus PR8 (H1N1) infection in mice. After 7 weeks of this specialized diet, mice were challenged with PR8 (H1N1) intranasally, and clinical signs were observed until 14 dpi. Humane endpoint criterion was <70% of initial body weight. All mice in the standard diet group died by 9 dpi, and only one mouse in the 25(OH)D**_3_**-fed group survived after 9 dpi (Figure 3a). The survival rates of the two groups were compared to the rate determined in a log rank (Mantel–Cox) test, which revealed no significant difference (*p* > 0.05). The body weight of mice in the 25(OH)D**_3_**-fed group were significantly higher than those of mice in the standard diet group at 5 dpi (Figure 3b).

To investigate whether 25(OH)D_3_ supplementation inhibits viral replication in mice at two different acute phases, the viral titer in the mouse lungs was measured by the plaque assay at 3 or 5 dpi. The viral titers in 25(OH)D_3_-fed mice at 3 dpi were significantly lower than those in the standardly fed group, but no significant difference was observed between the two groups at 5 dpi (Figure 4).

### 3.4. Production of Anti-Inflammatory and Proinflammatory Cytokines after Viral Challenge in Mice with 25(OH)D_3_ Supplementation

The amounts of anti-inflammatory and proinflammatory cytokines in each group were quantified. The levels of the anti-inflammatory cytokine IL-10 and of the proinflammatory cytokines IL-5, TNF-α, IFN-γ, and GM-CSF increased after viral infection in both groups, regardless of 25(OH)D_3_ intake. In addition, the proinflammatory cytokines IL-2 and IL-12p70 decreased after infection in both groups (Figure 5a,b). The levels of anti-inflammatory IL-4 and IL-10 were not significantly different in the lungs of 25(OH)D_3_-fed mice and standardly fed mice (Figure 5a). The levels of IFN-γ and IL-5, however, significantly decreased in the 25(OH)D_3_-fed group after viral challenge at 3 and 5 dpi, respectively (Figure 5b). By contrast, TNF-α production increased significantly in the 25(OH)D_3_-fed group at 5 dpi. In addition to these significant differences, a trend toward suppression of the levels of the proinflammatory cytokines TNF-α, GM-CSF, and IL-12p70 in the 25(OH)D_3_-fed group was observed at 3 dpi, and IL-2 production tended to be higher in the 25(OH)D_3_-fed group than in the standardly fed group at 3 dpi (0.05 < *p* < 0.1).

## 4. Discussions

Previous studies have suggested the use of dietary supplements with nonspecific immunostimulatory effects as a countermeasure against influenza virus infection [20]. Influenza is common in winter months, when the levels of 25(OH)D in the blood generally decrease. A correlation between influenza virus infection and deficiency of vitamin D has been previously observed [21]. The immunostimulatory effect of orally administered vitamin D in influenza patients has been clinically tested in humans, especially in infants, and a rapid abatement of fever and influenza A virus titers has been confirmed during the administration of high doses of vitamin D [22]; however, vitamin D consumption was not effective in elderly patients [12]. Hence, animal studies that allow investigation at the time of influenza virus infection are required. The compound 25(OH)D is metabolized from vitamin D in the liver and is better absorbed than vitamin D in the body [13]. After 25(OH)D circulates in the blood, it is metabolized to 1,25(OH)_2_D in the kidney as needed [23]. In a mouse model, suppression of inflammatory cytokines induced by H9N2 influenza virus infection by the intraperitoneal administration of 1,25(OH)_2_D after virus inoculation has been confirmed [24]. The half-life of 25(OH)D_3_ is relatively long (15 days), but the half-life of 1,25(OH)_2_D is only 15 h [14]. Therefore, we reasoned that long-term administration of 25(OH)D_3_ may mitigate the clinical manifestation of infectious diseases prophylactically. This study is the first to indicate a prophylactic effect against influenza virus infection in mice of the pre-administration of vitamin D metabolite 25(OH)D_3_ (125 ppm for 7 weeks).

The level of vitamin D not causing adverse effects in humans is 250 µg/day [25], and the adverse effects resulting from excessive consumption of vitamin D are renal insufficiency, neurologic manifestations, and pathologic calcification due to hypercalcemia [26]. Levels of 25(OH)D_3_ corresponding to 800–1000 ng/mL in mouse serum also showed adverse effects such as weight loss and hypercalcemia [27]. As shown in Figure 2, the blood level of 25(OH)D_3_ after 7 weeks of supplemental diet was 40–50 ng/mL, a concentration recommended as the minimum effective requirement against viral respiratory infections on the basis of observational studies [28]. Indeed, no adverse effects (e.g., weight loss, neurologic manifestations, or calcification of the organs), as described previously for 25(OH)D_3_-fed mice [29], were observed in this study. After 4 and 7 weeks of the supplemented diet, the plasma levels of 25(OH)D_3_ in 25(OH)D_3_-fed mice were significantly higher compared with those in standardly fed mice. In particular, the level of 24,25(OH)_2_D_3_ in 25(OH)D_3_-fed mice was higher than the level of 25(OH)D_3_ at 4 and 7 weeks, although the plasma level of 24,25(OH)_2_D_3_ is usually nearly equal to that of 25(OH)D_3_ in mice [27], suggesting that the uptake of 25(OH)D_3_ into the body was accelerated, and excessive 24,25(OH)_2_D_3_, a metabolite of 25(OH)D_3_, was also detected in the blood [16]. These results suggested that the dosage (125 ppm) and duration (7 weeks) of 25(OH)D_3_ administration secured a blood concentration of 25(OH)D_3_ sufficient to reduce the clinical manifestation of infectious diseases.

The influenza A virus PR8 (H1N1) strain is lethal in mice, who succumb within 8 dpi in the presence of a concentration of 10 MLD_50_ [17]. Influenza onset correlates with inflammatory cytokine expression, and neuroinflammation due to influenza virus infection releases inflammatory cytokines and can cause lung tissue damage [30]. Dysregulation of inflammatory cytokines in mice early in viral infection induces the overexpression of caspases and other proteolytic enzymes and plays a role in lung injury and lethality [31]. However, appropriate suppression of the expression of inflammatory cytokines may be able to suppress lethality caused by the influenza virus [32]. In a human trial, supplementation of vitamin D reduced the production of IL-5 after influenza virus infection [33]. The effects of 1,25(OH)_2_D include the repression of the transcription of T cell mRNA and the suppression of the production of IFN-γ [33]. After infection, the proinflammatory cytokines IFN-γ [33] and IL-5 [34] were reduced by the administration of 25(OH)D_3_. Anti-inflammatory cytokines suppress the immunopathological tissue damage caused by the production of inflammatory cytokines [35]. Consistent with previous reports, the levels of IL-5 and IFN-γ were suppressed in the 25(OH)D_3_-fed group. Notably, our study showed significant suppression of virus titers in the 25(OH)D_3_-fed group after 3 dpi, and this result supports a previous study that the growth of the H5N1 influenza virus in lungs was reduced by the administration of 1,25(OH)_2_D in mice [36]. These results suggest that long-term administration of 25(OH)D_3_ prior to virus inoculation suppressed the production of proinflammatory cytokines, resulting in reduced lung damage caused by virus infection and suppressed viral replication. In a previous report, the administration of 1,25(OH)_2_D to mice significantly inhibited the production of TNF-α after influenza virus infection [24]. However, in our study, a trend toward the suppression of TNF-α and a significant increase in TNF-α were observed in the 25(OH)D_3_-fed group after 3 and 5 dpi, respectively. In this study, we did not measure IL-6 levels, but there is a report that the administration of vitamin D could suppress the production of IL-6, which promotes the differentiation of B cells [12]. Therefore, the complex effects of cytokines must be elucidated in the future.

In conclusion, the present study showed that 25(OH)D_3_ supplementation adequately alleviated the clinical manifestations of influenza virus infection by suppressing virus replication and inflammation in a mouse model. In this result, there was no significant difference in the mortality rate of influenza virus infection following administration of 25(OH)D_3_, but we could confirm a tendency of reduced mortality rate. In other words, as a previous research showed [37], it is expected that the administration of vitamin D and its metabolites will reduce the mortality rate in an influenza pandemic. This evidence strongly supports the role of vitamin D in reducing the risk of respiratory viral diseases in humans, not only for influenza but also for the newly emerged coronavirus disease 2019 (COVID-19) pandemic [38]. Notably, however, 25(OH)D_3_ supplementation had little effect on mortality against influenza virus infection. In the future, the use of 25(OH)D_3_ combined with a vaccine [39] may synergistically reduce the mortality of influenza virus infection.

## Figures and Tables

**Figure 1 nutrients-12-02000-f001:**
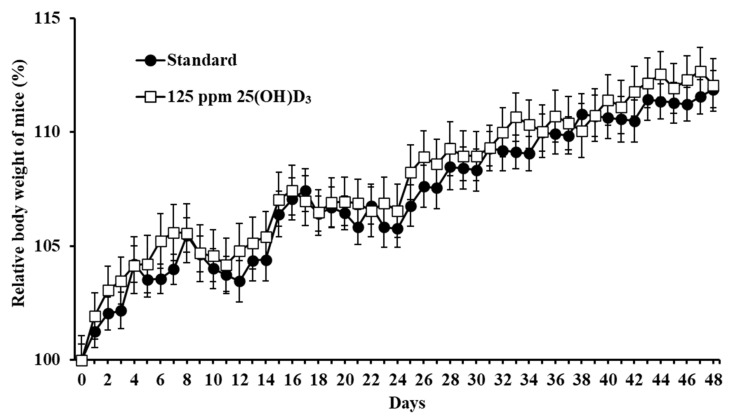
Changes of mouse body weight during dietary supplementation with 25(OH)D**_3_** for 7 weeks. Mice were divided into two groups and fed a standard diet or a 125 ppm 25(OH)D**_3_**-containing diet (*n* = 18 mice/group). The relative body weight of each mouse was measured for 7 weeks and compared with the primary body weight at day 0.

**Figure 2 nutrients-12-02000-f002:**
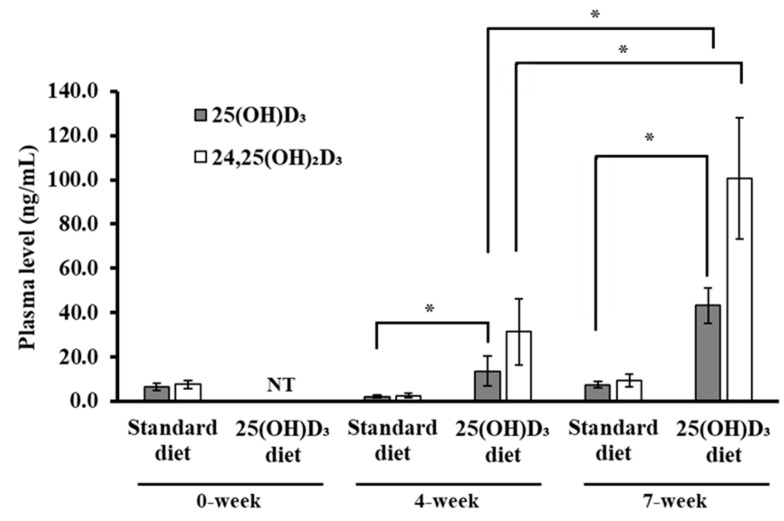
Plasma levels of 25(OH)D_3_ and 24,25(OH)_2_D_3_ in mice. Mice were fed a standard or a 125 ppm 25(OH)D_3_-supplemented diet for 7 weeks. The levels of 25(OH)D_3_ and 24,25(OH)_2_D_3_ were measured in each group (6 mice/group) at weeks 0, 4, and 7. Blood concentrations of 25(OH)D_3_ and 24,25(OH)_2_D_3_ in the 25(OH)D_3_-fed group at week 0 were not tested (NT). *, significant difference (*p* < 0.05).

**Figure 3 nutrients-12-02000-f003:**
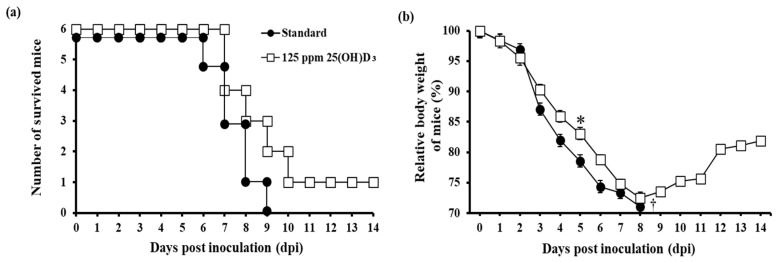
Survival and body weight of mice challenged with PR8 (H1N1) (*n* = 6 mice/group). (**a**) Number of surviving mice over time. (**b**) Relative body weight of mice after virus challenge. *, significant difference (*p* < 0.05); †, all mice deceased.

**Figure 4 nutrients-12-02000-f004:**
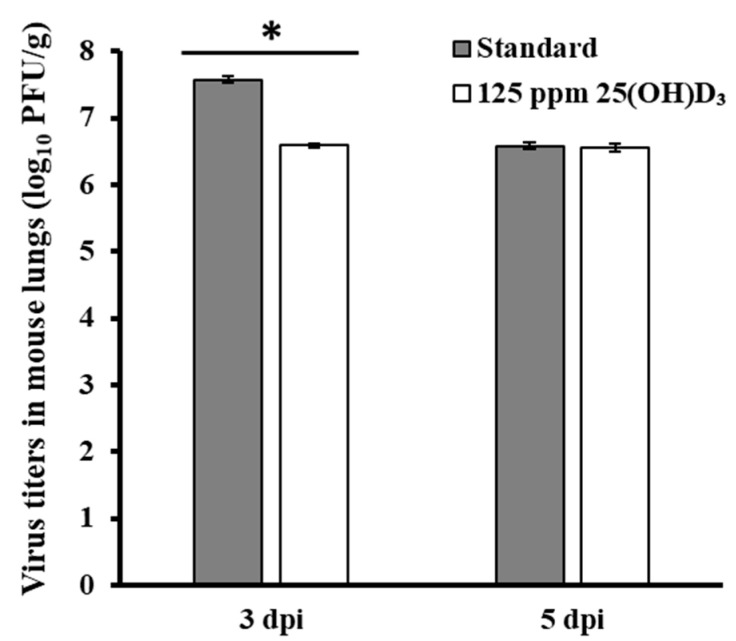
Virus recovery from the lungs of mice challenged with PR8 (H1N1) at 3 or 5 dpi. Mice were fed a standard diet or a diet supplemented with 125 ppm 25(OH)D_3_ for 7 weeks and then challenged with PR8 (H1N1) intranasally. The viral titers in the lung homogenates at 3 or 5 dpi were measured (PFU/g) for each group (*n* = 6 mice/group). *, significant difference (*p* < 0.05).

**Figure 5 nutrients-12-02000-f005:**
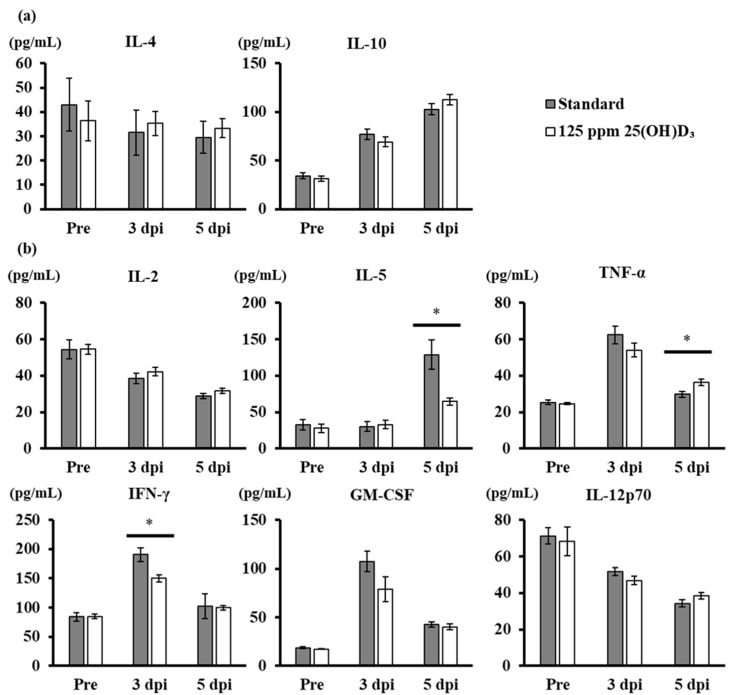
Levels of anti-inflammatory and proinflammatory cytokines before and after viral challenge (*n* = 6 mice/group). Mice were fed a standard diet or a diet supplemented with 125 ppm 25(OH)D_3_ for 7 weeks and then challenged with PR8 (H1N1) intranasally. (**a**) Concentrations of anti-inflammatory cytokines IL-4 and IL-10. (**b**) Concentrations of proinflammatory cytokines IL-2, IL-5, TNF-α, IFN-γ, GM-CSF, and IL-12p70. *, significant difference (*p* < 0.05).

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
