# Peer review of "Oral Supplementation of the Vitamin D Metabolite 25(OH)D3 Against Influenza Virus Infection in Mice"

_nutrients, 2020, doi:10.3390/nu12072000_

Round 1

Reviewer 1 Report

The study seems to be well conducted, but the finding that vitamin D had no effect on mortality may not be applicable to humans. In this paper, it was suggested that vitamin D reduced risk of death after developing pandemic influenza due to reducing risk of ensuing pneumonia. Do mice develop pneumonia?

The possible roles of solar ultraviolet-B radiation and vitamin D in reducing case-fatality rates from the 1918-1919 influenza pandemic in the United States.

Grant WB, Giovannucci E.Dermatoendocrinol. 2009 Jul;1(4):215-9. 

Why was IL-6 not measured?

What about cathelicidin and defensins affected by vitamin D?

Table 1 in Ref. 28 in the manuscript reviewed vitamin D supplementation trials regarding influenza such as this one. The results in that table might be summarized in the manuscript.

Randomized trial of vitamin D supplementation to prevent seasonal influenza A in schoolchildren.

Urashima M, Segawa T, Okazaki M, Kurihara M, Wada Y, Ida H.Am J Clin Nutr. 2010 May;91(5):1255-60.

This paper might be discussed

Identification of amitriptyline HCl, flavin adenine dinucleotide, azacitidine and calcitriol as repurposing drugs for influenza A H5N1 virus-induced lung injury.

Huang F, Zhang C, Liu Q, Zhao Y, Zhang Y, Qin Y, Li X, Li C, Zhou C, Jin N, Jiang C.PLoS Pathog. 2020 Mar 16;16(3):e1008341. 

Author Response

Reviewer #1

Comment 1. The study seems to be well conducted, but the finding that vitamin D had no effect on mortality may not be applicable to humans. In this paper, it was suggested that vitamin D reduced risk of death after developing pandemic influenza due to reducing risk of ensuing pneumonia. Do mice develop pneumonia?

The possible roles of solar ultraviolet-B radiation and vitamin D in reducing case-fatality rates from the 1918-1919 influenza pandemic in the United States (Grant WB, Giovannucci E.Dermatoendocrinol. 2009 Jul;1(4):215-9. )

Answer: Thank you for your constructive comment. The strain of influenza A virus in this study was A/Puerto Rico/8/1934 (H1N1). In the previous report, it was indicated that the mice inoculated this virus strain developed pneumonia and lead to the death.

Fukushi et al., Serial Histopathological Examination of the Lungs of Mice Infected with Influenza A Virus PR8 Strain. PLoS One, 2011;6: doi: 10.1371/journal.pone.0021207  

Therefore, we believe that the results of the present study in mice can be extrapolated to the human effect. In our statistical analysis (p<0.05), there was no significant difference in the mortality rate of influenza virus infection after administration of 25(OH)D3, but we could confirm the tendency to suppress the mortality of mice in our study. Hence, similar to the previous research, it is expected that administration of vitamin D and its metabolites will reduce the mortality rate in the new pandemic in humans.

Modification: We added the sentence about the previous reports at P. 8, L. 257-261. In addition, the reference was cited in the text as No. 37.

Comment 2. Why was IL-6 not measured?

Answer: In this study, we measured proinflammatory cytokines IFN-γ and TNF-α with high priority to investigate the inflammation in the lungs. Actually, we could not measure the IL-6 in our experiment using Bio-Plex Pro mouse cytokine Th1/Th2 assay (Bio-Rad, Hercules, CA, USA). On the other hand, there is a previous report that administration of vitamin D could suppress the production of IL-6, which is important the differentiation of B cells.

Goncalves-Mendes et al., Impact of Vitamin D Supplementation on Influenza Vaccine Response and Immune Functions in Deficient Elderly Persons: A Randomized Placebo-Controlled Trial. Front Immunol. 2019 ,10:65, doi: 10.3389/fimmu.2019.00065.

Taken together, we conclude that elucidating the regulatory mechanisms of cytokines, including IL-6, is a topic for future research.

Modification: We added the sentence about the previous reports at P. 8, L. 252-253. In addition, the reference was cited in the text as No. 12.

Comment 3. What about cathelicidin and defensins affected by vitamin D?

Answer: Thank you for pointing out. In humans, administration of vitamin D or 1,25(OH)2D could enhance the production of antimicrobial peptides. But previous report revealed that administration of 1,25(OH)2D did not induce the production of antibacterial peptides cathelicidin and defensins in mice.

Gombart et al., Human cathelicidin antimicrobial peptide (CAMP) gene is a direct target of the vitamin D receptor and is strongly up‐regulated in myeloid cells by 1,25‐dihydroxyvitamin D3, FASEB J. 2005:9:.doi: 10.1096/fj.04-3284com.

Therefore, we considered that administration of 25(OH)D3 does not enhance the production of cathelicidin and defensins in our study.

Modification: We had no modification.

Comment 4. Table 1 in Ref. 28 in the manuscript reviewed vitamin D supplementation trials regarding influenza such as this one. The results in that table might be summarized in the manuscript.

Randomized trial of vitamin D supplementation to prevent seasonal influenza A in schoolchildren. (Urashima M, Segawa T, Okazaki M, Kurihara M, Wada Y, Ida H.Am J Clin Nutr. 2010 May;91(5):1255-60.)

Answer: We agree with your comment. In the second paragraph of our discussion, we quoted this reference report as an example of a decrease in the incidence of influenza caused by vitamin D administration.

Modification: The reference was cited in the text as No.28 at P. 8, L. 221.

Comment 5. This paper might be discussed

Identification of amitriptyline HCl, flavin adenine dinucleotide, azacitidine and calcitriol as repurposing drugs for influenza A H5N1 virus-induced lung injury (Huang F, Zhang C, Liu Q, Zhao Y, Zhang Y, Qin Y, Li X, Li C, Zhou C, Jin N, Jiang C.PLoS Pathog. 2020 Mar 16;16(3):e1008341.)

Answer: Thank you for pointing out. This reference paper was cited in the 3rd paragraph of the discussion as an example in which the growth of the H5N1 influenza virus in lungs was reduced by the administration of 1,25(OH)2D.

Modification: We added the sentence about the referenced report at P. 8, L. 244-246. The reference was cited in the text as No. 36.

Reviewer 2 Report

This is a manuscript by Hayashi H et al on oral supplementation of Vitamin D metabolite against influenza infection in mice.
The manuscript is well written and designed. I have only a minor concern. At paragraph 3.4, lines 191-195. This meaning seems to me to fit better at the material and methods than results.

Author Response

Reviewer #2

Comment 1. The manuscript is well written and designed. I have only a minor concern. At paragraph 3.4, lines 191-195. This meaning seems to me to fit better at the material and methods than results.

Answer: Thank you very much for your suggestion. We deleted the sentence which you pointed out because the description is a repeat of Materials and Methods.

Modification: We deleted above sentence from P. 6, L. 177-178.

Reviewer 3 Report

This experiment shows the function of Vitamin D on flu infection with mouse model. This manuscript tests the possibility of long term treatment of mice with Vitamin D to prevent flu infection. This manuscript should be further strengthened by addressing a few concerns as follow:

  1. Do the authors do the repeat experiment for Figure 3. Could the authors show all the data ( included repeated data) in Figure 3.
  2. In Figure 4, the error bar of Viral titers are so tiny. Do the authors not show the real error bar by mistake?

Author Response

Reviewer #3

Comment 1. Do the authors do the repeat experiment for Figure 3. Could the authors show all the data (included repeated data) in Figure 3.

Answer: Thank you for pointing out. We consider it desirable to do the re-experiment and confirm reproducibility, but unfortunately, we do not have these results. However, the results shown in our paper are considered to be highly accurate because prior to the animal experiments to obtain the results shown in our paper, preliminary animal experiments were performed.

Modification: We had no modification.

Comment 2. In Figure 4, the error bar of Viral titers are so tiny. Do the authors not show the real error bar by mistake?

Answer: Thank you for your suggestion. Since there was not much difference in the virus titer among the individual mice, it was confirmed that the standard error was small. For reference, the virus titer in the lungs of each mouse is shown below.

All units are log10 PFU/g

<3 dpi>

Standard group: 7.5, 7.5, 7.5, 7.7, 7.6, 7.7

25(OH)D3-fed group: 5.6, 5.6, 5.5, 5.6, 5.7, 5.5

<5 dpi>

Standard group: 6.5, 6.7, 6.5, 6.7, 6.6, 6.5

25(OH)D3-fed group: 6.3, 6.7, 6.5, 6.7, 6.6, 6.5

However, after the statistical analysis was performed again, the numbers of standard errors were corrected as follows.

<3 dpi>

Standard group: Previous manuscript: 0.03, Revised manuscript: 0.04

25(OH)D3-fed group: Previous manuscript: 0.02, Revised manuscript: 0.03

<5 dpi>

Standard group: Previous manuscript: 0.04, Revised manuscript: 0.05

25(OH)D3-fed group: Previous manuscript: 0.05, Revised manuscript: 0.06

Modification: We replaced Figure 4.

Round 2

Reviewer 3 Report

The repeated experiment for Fig. 3 is essential.